# Using cancer risk algorithms to improve risk estimates and referral decisions

Olga Kostopoulou [1✉], Kavleen Arora[1] & Bence Pálfi[1]

## Abstract

**Background** Cancer risk algorithms were introduced to clinical practice in the last decade, but they remain underused. We investigated whether General Practitioners (GPs) change their referral decisions in response to an unnamed algorithm, if decisions improve, and if changing decisions depends on having information about the algorithm and on whether GPs overestimated or underestimated risk.

**Methods** 157 UK GPs were presented with 20 vignettes describing patients with possible colorectal cancer symptoms. GPs gave their risk estimates and inclination to refer. They then saw the risk score of an unnamed algorithm and could update their responses. Half of the sample was given information about the algorithm's derivation, validation, and accuracy. At the end, we measured their algorithm disposition. We analysed the data using multilevel regressions with random intercepts by GP and vignette.

**Results** We find that, after receiving the algorithm's estimate, GPs' inclination to refer changes 26% of the time and their decisions switch entirely 3% of the time. Decisions become more consistent with the NICE 3% referral threshold (OR 1.45 [1.27, 1.65], p < .001). The algorithm's impact is greatest when GPs have underestimated risk. Information about the algorithm does not have a discernible effect on decisions but it results in a more positive GP disposition towards the algorithm. GPs' risk estimates become better calibrated over time, i.e., move closer to the algorithm.

**Conclusions** Cancer risk algorithms have the potential to improve cancer referral decisions. Their use as learning tools to improve risk estimates is promising and should be further investigated.

### Plain language summary

Cancer risk algorithms are statistical formulae that calculate the probability that a patient presenting with certain symptoms has cancer. Their aim is to aid primary care physicians when deciding whether a patient should be seen by an oncologist urgently. We presented 157 UK primary care physicians with 20 descriptions of hypothetical patients at varying degrees of risk and asked how likely they would be to refer them to an oncologist for suspected cancer urgently. We then showed them the risk (probability of these patients having cancer), as calculated by an algorithm. The algorithm changed the physicians' inclination to refer the patients 26% of the time. Decisions improved overall. We propose such algorithms be used to aid cancer referral decisions, and to train doctors making decisions about which patients should be seen by oncologists urgently.

[1] Imperial College London, Department of Surgery & Cancer, London, UK.   ✉email: o.kostopoulou@imperial.ac.uk

mproving cancer outcomes in England is a national priority. In 2018, 55% of cancers were diagnosed at stages 1 and 2[1]. NHS England aims to raise this to 75% by 2028 by improving the early diagnosis of cancer[2]. General Practitioners (GPs) can use the 2-week-wait (2WW) referral pathway if they suspect cancer; the patient is then seen by a specialist within a target of two weeks. It was recently demonstrated that the 2WW pathway is effective in improving cancer outcomes: higher use of the pathway was associated with lower mortality for common cancers and lower odds of late-stage diagnosis[3]. However, large variability between practices in their use of the 2WW referral pathway[4] means that it may not fulfil its full potential. The variability has partly been explained by the organisation of the local health services[5] and partly by GP decision making[6,7]. Discriminating patients who should be referred on the 2WW pathway from those who do not need to is difficult, especially where early cancers present with vague, non-specific symptoms that could easily be attributed to other conditions[8]. Using cancer risk calculators could improve cancer referral decision making[9], by helping GPs identify at-risk patients and, thus, reduce diagnostic delay, while reassuring them about low-risk patients who do not require referral, and thus avoid overloading the healthcare system.

Cancer risk calculators are algorithms that calculate the probability that a patient with symptoms has a current, undiagnosed cancer. QCancer[10] and RAT[11] are two established cancer risk calculators which have been integrated with the electronic health record in some parts of UK primary care. Studies of the implementation of cancer risk calculators in clinical practice have had mixed results: a cohort study found an increase in the number of investigations ordered and cancers diagnosed after RATs were provided to primary care clinics[12]; a cluster randomised trial found no impact of GP education resources, which included RATs, on time to diagnosis[13]; and a qualitative study of GPs doing simulated consultations suggested distrust of QCancer when it conflicted with clinical judgement[14]. Indeed, despite these tools being available in primary care for almost a decade and their potential to improve the earlier diagnosis of cancer, they remain an underused resource[15].

The study reported here is the first in a planned series of studies aiming to investigate how cancer risk algorithms influence clinical risk assessments and referral decisions, and to identify ways to optimise their introduction and presentation. The study involved GPs responding to a series of clinical vignettes online. We investigated whether GPs change their referral decisions in response to an unnamed algorithm, if decisions improve, and what factors influence decision change. The factors that we investigated were the provision of information about the algorithm, and the position of the GPs' initial risk estimates in relation to the algorithm, i.e., underestimation vs. overestimation of risk. GPs are not routinely informed about how algorithms that are introduced in their electronic health record have been elicited and validated and how accurate they are[16]. It is plausible to expect that such information would improve trust in the algorithm and lead to greater willingness to follow its advice and integrate its probabilities into one's own risk assessment and referral decisions. We also expected that GPs would err on the side of caution, putting more importance on misses than false-positive referrals, and thus be less willing to change a referral decision when the algorithm suggested that the patient's risk was lower (vs. higher) than what they had initially thought. Furthermore, we investigated GPs' disposition towards the algorithm and associations with GP demographics, prior attitudes towards cancer risk calculators, and decision confidence.

Only one of our hypotheses was confirmed: GPs were indeed more likely to change a referral decision if they had initially underestimated vs. overestimated risk. Having information about the algorithm's derivation, validation and accuracy did not impact decisions. We measured a statistically significant improvement of decisions vis-à-vis the NICE 3% referral threshold. Finally, we observed that GPs' risk estimates moved closer to the algorithm as the algorithm's estimates were repeatedly presented to them over the series of vignettes. This is encouraging, as it suggests that such algorithms could be used to train clinicians to estimate cancer risk better.

## Methods

**Sample size.** We powered the study to detect a small effect ($f^2 = 0.02$) of the algorithm on referral decisions with alpha of 5% and power of 95% in a multiple linear regression. The G*Power software (v. 3.1.9.4) estimated that we would need at least 863 responses. To account for data clustering (each GP responding to 20 vignettes), we adjusted this number by the Design Effect (DE)[17]. This is calculated using the formula $DE = 1 + (n-1)*ICC$, where n is the cluster size (the 20 vignettes), and ICC is the intra-class correlation. We estimated the ICC from pilot data to be 0.088. Thus, DE = 2.68. We adjusted the number of participants required by multiplying the 863 required responses with the DE and dividing by the cluster size: $(863*2.68)/20 = 116$. Thus, we estimated that we needed to recruit a minimum of 116 GPs.

**Materials.** We prepared 23 clinical vignettes, each having a different combination of risk factors, symptoms and signs related to colorectal cancer. To prepare the vignettes, we used QCancer (https://qcancer.org), which is publicly available, as is its underlying computer code. We selected from the range of risk factors and symptoms that QCancer uses and employed them in different combinations, aiming for clinical plausibility and a wide range of risk across vignettes. Vignette risk ranged from 0.58% to 57.23% (mean 14.10%, SD 18.97, median 4.18). As well as creating some new vignettes, the majority were modified from those used in a previous study by the lead author[7].

Each vignette described a hypothetical patient presenting in general practice. All vignettes started with a list of demographics and risk factors (name, sex, age, BMI, smoking and alcohol intake), followed by the presenting problem. One or more of the relevant risk factors and symptoms in QCancer (type 2 diabetes, family history of gastrointestinal cancer, weight loss, appetite loss, abdominal pain, rectal bleeding, change in bowel habit, constipation, and anaemia) were incorporated into the description. All the vignettes are presented in the Supplementary Methods.

Three of the vignettes were used for familiarisation purposes and no data were collected. The remaining twenty were split into two sets of ten to be completed on two different days to minimise fatigue. We made sure that the range, median, mean and standard deviation of risk estimates were almost identical in the two sets. We also counterbalanced the sets across participants, so that each set was completed first and second an equal number of times. All materials were presented online on the Qualtrics platform (qualtrics.com).

**Procedure.** Study approval was provided by the Health Research Authority (HRA) and Health & Care Research Wales (HCRW), REC reference 20/HRA/2418. An invitation email was sent to the 400 GPs in our database—a database of e-mails compiled by the lead author and consisting of participants in previous studies, all currently practising in England. The invitation email included a brief description of the study and outlined the benefits of participation: remuneration of £60, a completion certificate, and personalised feedback to use as evidence of continuous professional development (CPD) for their appraisal portfolio. Those

interested in participating could follow a link in the email, which took them to an expression-of-interest form, where they could enter their NHS email address and GP practice code.

After participants accessed the study site, they read an information sheet and provided consent online. They then completed demographics questions (age, gender, GP or GP trainee, year of GP qualification, and number of clinical sessions per week), and answered the following questions: *"In general, how confident do you feel when assessing patients with symptoms that might indicate cancer?"* (I always feel confident/I feel confident most of the time/I feel confident sometimes/I seldom feel confident). *"Are you aware of any cancer risk algorithms that are being used in clinical practice to calculate a patient's current risk of cancer (aka 'cancer risk calculators')?"* (Yes/No). If they answered "yes", they were then asked: *"Are they available in the electronic health record that you use in your practice?"* (Yes/No). If they answered "yes", they were then asked to indicate which cancer risk algorithms were available in their practice, and they could choose one or more of the following options: RAT, QCancer, C the Signs, and Other. They were then asked how often they used these cancer risk algorithms (always/sometimes/never). Finally, all participants were asked to rate their attitude towards cancer risk calculators on a scale from "Very negative" (1) to "Very positive" (9).

Half of the participants were randomly allocated to receive the following information about the study algorithm (Box 1):

Participants in the algorithm information group then responded to three questions gauging understanding and trust. Specifically, they were asked if the description of the algorithm made sense to them (Yes/No), if they would trust this algorithm's estimates (Definitely yes/Probably yes/ Probably not/Definitely not), and if they would like to have an algorithm like this in their clinical practice (Definitely yes/Probably yes/Probably not/ Definitely not).

All participants were then presented with the three practice vignettes in a random order. No data were collected at this stage and participants were informed of this. The aim of the practice vignettes was to familiarise participants with the task and help them calibrate their risk estimates, since GPs do not provide explicit cancer risk estimates on a routine basis. For this purpose, the practice vignettes represented three levels of risk of undiagnosed colorectal cancer: low (1%), medium (6%) and high (40%).

The ten vignettes of the first set then followed in a random order. The procedure was exactly the same for all the vignettes,

---

**Box 1**

The algorithm aims to be used as a decision aid, to support 2WW cancer referral decisions. It is not intended to determine those decisions.

The algorithm was derived from a large cohort study of 2.5 million patients in the UK. They used data in the primary care record of cancer patients to estimate associations between risk factors, symptoms/signs and a subsequent cancer diagnosis.

The algorithm estimates the probability that a patient has colorectal cancer, given his/her risk factors and presenting symptoms/signs; in other words, how many people out of 100 with the same risk factors and presenting symptoms/signs are likely to have colorectal cancer.

A study that validated the algorithm on another large cohort of patients, a proportion of whom had colorectal cancer, found that the algorithm performed very well: it discriminated correctly between cancer and non-cancer patients approximately 90% of the time (i.e., produced higher risk estimates for cancer than non-cancer patients).

---

including the practice vignettes. Specifically, each vignette was followed by three questions:

1. *"Out of 100 patients with the same risk factors and symptoms as this patient, how many, in your clinical judgement, are likely to have colorectal cancer? Please type in a whole number between 0 and 100."* Responses could be typed in a box below the question.

2. *"What is the narrowest range which you are almost certain contains your estimate above? Enter the lower and upper limits in the boxes below. Make sure that your estimate falls within this range."*
   Respondents filled in the lower and upper limits in the following sentence: *"I am almost certain that out of **100** patients like this one, between <lower limit> and <upper limit> are likely to have colorectal cancer as yet undiagnosed."*

3. *"How likely is it that you would refer this patient on the 2WW pathway for suspected cancer **at this consultation**?"* Responses were given on a rating scale: 1 (highly Unlikely), 2 (Unlikely), 3 (Uncertain), 4 (Likely), 5 (Highly likely).

NB. Words in bold or italics appeared on the screen as they appear above.

After these three questions were answered, the vignette was presented again, this time with the algorithmic estimate: *"The algorithm estimates that <number> out of 100 patients presenting like this is/are likely to have colorectal cancer. Your estimate was <number> out of 100 (lower limit <number>, upper limit <number>). If you wish to revise your initial estimates, please do so below. If you wish to stick with your initial estimates, please re-enter them below."* Participants were then invited to answer the same three questions as before.

Following completion of the first 10 vignettes, participants had the opportunity to give feedback on any aspect of the study in free text. Twenty-four hours after completing the first set of 10 vignettes, participants were automatically sent a link to the second set. The procedure in the second study session was the same as in the first session. Participants who had received information about the algorithm in the first session were presented with it again at the start of the second session. After completing the second set of vignettes, all participants completed the Algorithm Disposition Questionnaire (ADQ). The ADQ consisted of seven statements:

1. I found the algorithm's risk estimates helpful.
2. I think that the algorithm's estimates were accurate.
3. I felt irritated when receiving the algorithm's estimates.
4. I was happy to receive the algorithm's estimates.
5. I was frustrated when receiving the algorithm's estimates.
6. I felt more confident in my referral decisions, having received the algorithm's estimates.
7. I feel appreciative having access to the algorithm's estimates.

Respondents indicated their agreement with each statement on 7-point scales: 1 (strongly disagree), 2 (disagree), 3 (slightly disagree), 4 (neither disagree nor agree), 5 (slightly agree), 6 (agree), 7 (strongly agree). Statements 3 and 5 were reverse-scored. Finally, all participants were given the opportunity to comment on any aspect of the study, if they wished. Data collection took place between 27th June 2020 and 23rd September 2020—dates of first and last study completion.

**Statistics and reproducibility**. We aimed to measure the impact of our manipulation (algorithm information provided vs. not provided) and GPs' over- vs. underestimation of risk on referral

decisions. To this end, we created several variables, which we subsequently used in regression analyses.

**Creation of variables**. We created a dichotomous variable denoting the position of the GPs' initial (i.e., pre-algorithm) risk estimates in relation to QCancer: overestimation (1) vs. underestimation (0). We excluded responses where intuitive estimates matched QCancer. To measure changes in risk estimates, we subtracted the final from the initial estimate, and signed the difference so that positive values indicated changes consistent with the algorithm (the final estimate was closer to the algorithm than the initial estimate) and negative values indicated changes inconsistent with the algorithm. Similarly, to measure changes in referral inclination, we subtracted the final from the initial response on the 1-5 scale and signed the difference so that positive values indicated changes consistent with the algorithm and negative values indicated changes inconsistent with the algorithm. For example, if the algorithm estimated a higher risk than the GP, who subsequently gave a higher value on the response scale, the raw difference of the two response values on the scale would be negative but the adjusted would be positive. We also created a simpler, dichotomous variable for referral inclination, indicating whether respondents moved from one point on the response scale to another: change (1) vs. no change (0).

To determine whether referral decisions improved post-algorithm, we created two dichotomous variables: decision appropriateness (appropriate vs. not appropriate) and time of decision (pre- vs. post-algorithm). We defined decision appropriateness using the NICE risk threshold of 3% (https://www.nice.org.uk/guidance/ng12/evidence/full-guideline-pdf-2676000277). Therefore, if GPs indicated that they were either likely or highly likely to refer a vignette with QCancer risk score ≥3%, the decision was classed as appropriate. Similarly, if they indicated that they were either unlikely or highly unlikely to refer a vignette with QCancer risk score <3%, the decision was classed as appropriate. Otherwise, it was classed as inappropriate.

**Regression models**. All regression models were multilevel with random intercepts by GP and vignette, unless otherwise indicated. The regression tables are presented in Supplementary Note 1, in the sequence that they appear in the text. First, we ran two empty regression models, one for risk estimate changes and the other for changes in referral inclination to measure the impact of the algorithm on these two behavioural measures. To measure whether changes in risk estimates were associated with changes in referral inclination, we regressed inclination changes on risk estimate changes. We repeated the analysis as a logistic regression, using the simpler, dichotomous variable for inclination changes (change vs. no change). We then regressed each referral inclination variable on the two predictors of interest (algorithm information and position of GPs' initial estimates vis-à-vis QCancer). We also explored the contribution of other variables by subsequently adding them to these two regression models in a single step: GP demographics (gender, years in general practice); confidence when assessing patients with symptoms that might indicate cancer; and general attitude towards cancer risk calculators.

Using logistic regression, we regressed decision appropriateness on time of decision. In one analysis, uncertain decisions (i.e., those at the midpoint of the decision scale) were classed as inappropriate. We then repeated the analysis excluding uncertain decisions from the calculations. Finally, we explored whether any learning had taken place as a result of the QCancer score repeatedly presented after each vignette, by measuring whether

GPs' initial risk estimates improved over time, i.e., moved closer to QCancer. We defined improvement as a reduction in the difference between GPs' initial risk estimates and QCancer. We used the absolute values of this difference to avoid situations where overestimation and underestimation cancelled each other out. We regressed this absolute difference on study session (1st vs. 2nd); in a separate model, we regressed it on vignette order (1–20).

Finally, we explored predictors of algorithm disposition. Using simple linear regression, we regressed participants' score on the Algorithm Disposition Questionnaire (ADQ score) on GP demographics (gender, years in general practice), confidence when assessing patients with symptoms that might indicate cancer, general attitude towards cancer risk calculators, and algorithm information (present vs. absent).

We calculated the explained variance for each regression model using the r.squaredGLMM function of the MuMIn R package[18], which is based on the work of Nakagawa and colleagues[19]. We report both the marginal and conditional $R^2$ (Supplementary Note 1). The marginal $R^2$ indicates the explained variance by the fixed factors, and the conditional $R^2$ indicates the variance explained by the whole model including the random effects. All analyses were conducted using Stata 17.0 and were confirmed in R (version 4.0.3). The dataset can be found in Supplementary Data 1.

**Reporting summary**. Further information on research design is available in the Nature Research Reporting Summary linked to this article.

## Results
We recruited 150 fully qualified GPs and 7 GP trainees from a total of 124 GP practices (primary care clinics) across England. The number of GPs working in the same practice ranged from 1 to 5 (Median 1). Most practices (101/124, 81%) were represented by one GP in the sample. The mean age of the sample was 44 years (SD 8.7) and 53.5% of the participants were female. Average experience was 14 years since qualification (SD 9, Median 12). Three GPs answered this question giving one or two-digit numbers rather than a year and were thus excluded from the calculation of experience. Most participants indicated that they were confident most of the time when assessing patients with symptoms that might indicate cancer (2380/3140, 76%), while a substantial minority were confident only some of the time (620/3140, 20%).

**Awareness, frequency of use and attitudes towards cancer risk calculators**. Most participants were aware of cancer risk calculators (108/157, 69%). However, only 47 GPs (29.9%) had cancer risk calculators available in the electronic health record at their practice, with QCancer being the most common (Table 1). The sample reported a generally positive attitude towards cancer risk calculators, with a mean of 5.99 on the response scale from very negative (1) to very positive (9) (SD 1.54). Table 1 presents the number of GPs who indicated that a specific type of cancer risk calculator was available in the electronic health record at their practice, and their attitude towards these calculators. Table 2 shows that where a calculator was known to be available, half of the respondents used it sometimes and a large minority (40%) never used it. Thus, out of a total of 157 participants, only 28 (18%) were actively using a cancer risk calculator either sometimes or always.

We provided half of our sample (80/157, 51%) with information about the algorithm at the start of the study. Most reported that the information made sense (78/80, 98%), that they

**Table 1 Access to and attitude towards cancer risk calculators.**

| Type of cancer risk calculator available at the practice | GPs with access to a cancer risk calculator at their practice | Attitude towards risk calculators* (mean, SD) |
|---|---|---|
| Qcancer | 26 (55%) | 5.08 (1.72) |
| C the Signs | 7 (15%) | 6.43 (1.27) |
| Qcancer & C the Signs | 9 (19%) | 6.11 (2.09) |
| Qcancer & RAT | 2 (4%) | 7 (2.83) |
| Other | 3 (6%) | 4 (2.65) |
| Total | 47 (100%) | 5.48 (1.88) |

Numbers (%) of GPs who indicated that they had access to one or more cancer risk calculators at their practice and their attitude towards them, presented by type of cancer risk calculator available.
* "In general, how do you feel about having cancer risk calculators in clinical practice?" Response scale: "very negative" [1] to "very positive" [9].

**Table 2 Frequency of use and attitudes towards cancer risk calculators.**

| Frequency of use of cancer risk calculators | GPs with access to a cancer risk calculator at their practice | Attitude towards cancer risk calculators (mean, SD) |
|---|---|---|
| Always | 4 (9%) | 8.25 (0.96) |
| Sometimes | 24 (51%) | 5.75 (1.70) |
| Never | 19 (40%) | 4.58 (1.64) |
| Total | 47 (100%) | 5.48 (1.88) |

Frequency of cancer risk calculator use, where they were known to be available, and GPs' attitude towards them.

**Table 3 Frequency of referral decisions pre- and post-algorithm.**

| Referral decisions | Pre-algorithm | Post-algorithm |
|---|---|---|
| Unlikely (1) or highly unlikely (2) | 545 (17.36%) | 637 (20.29%) |
| Uncertain (3) | 418 (30.67%) | 381 (32.42%) |
| Likely (4) or highly likely (5) | 2177 (69.33%) | 2122 (67.58%) |
| Total | 3140 (100%) | 3140 (100%) |

Decisions were measured on a 5-point scale ranging from 1 (highly unlikely) to 5 (highly likely), with a midpoint of 3 (uncertain).

would trust the algorithm's estimates ("definitely yes" or "probably yes": 72/80, 90%), and that they would like to have an algorithm like this in their clinical practice ("definitely yes" or "probably yes": 72/80, 90%).

**Referral decisions pre- and post-algorithm**. We collected a total of 3140 decisions about referral (157 participants responding to 20 vignettes). Table 3 summarises the moves on the decision response scale. Participants moved to a different point on the response scale after seeing the algorithm in a quarter of responses (808/3140, 26%). In 12% of these responses, they switched their decisions entirely (95/808): from referral to no-referral (from 4 or 5 to 1 or 2 on the response scale) in 47 responses; and from no-referral to referral (from 1 or 2 to 4 or 5 on the response scale) in 48 responses. In the remainder of responses (713/808), only the

inclination to refer changed, either increasing (398/808, 49%) or decreasing (315/808, 39%).

**Changes in risk estimates and referral inclination and their association**. Both risk estimates and referral inclination changed significantly after the algorithm was received; risk estimate changes: $b = 10.23$ [7.49, 12.97], $p < 0.001$; referral inclination changes: $b = 0.25$ [0.20, 0.31] $p < 0.001$ (Supplementary Note 1, Tables S1, S2). The regression coefficients indicate that GPs changed their estimates in accordance with the algorithm by 10.23% on average and moved on the 1–5 decision response scale by a quarter of a unit on average. We found a weak but statistically significant association between changes in risk estimates and changes in referral inclination ($b = 0.016$ [0.01, 0.02], $p < 0.001$, $f^2 = 0.09$) (Supplementary Note 1, Table S3a, b).

We observed risk overestimation (initial estimate > QCancer score) in 70% of responses (2211/3140), underestimation in 23% of responses (714/3140), while initial estimates matched QCancer scores exactly in 7% of responses (215/3140). We categorised changes in referral inclination as either towards referral, i.e., any increase in value on the 1–5 response scale, or away from referral, i.e., any reduction in value on the 1–5 response scale. Table 4 shows that where GPs became more inclined to refer, they had underestimated risk on average and increased their risk estimates after seeing the algorithm; where they became less inclined to refer, they had overestimated risk on average and reduced their risk estimates after seeing the algorithm.

**Impact of algorithm information and position of initial risk estimates**. Referral inclination changed in accordance with the algorithm more when risk was initially underestimated vs. overestimated ($b = 0.32$ [0.26, 0.39], $p < 0.001$, $f^2 = 0.03$); we detected no effect of algorithm information (Supplementary Note 1, Table S4a). When we used the dichotomous decision variable in the regression, we found that the odds of change almost tripled when risk was initially underestimated vs. overestimated ($OR = 2.96$ [2.14, 4.08], $p < 0.001$); the test of algorithm information was not significant (Supplementary Note 1, Table S4b). Thus, it appears that GPs were being cautious and less willing to change their initial estimates and referral decisions when the algorithm suggested that they had overestimated cancer risk than when it suggested that they had underestimated it. Some GPs acknowledged this in their written comments. For example:

GP 12635: *"Low likelihood on the algorithm doesn't really influence if I change my answers but a high likelihood does."*

GP 27092: *"I think when the algorithm supported my decision, I found it helpful but when I would have thought to refer but it gave a low estimate, I often ignored it… If my own risk assessment was low but the algorithm's was high, then I'd be more likely to err on the side of caution."*

We repeated these regression analyses adding GP demographics, confidence when assessing possible cancers, and general attitude towards cancer risk calculators. We detected a significant negative relationship between changes in referral inclination and confidence when assessing patients with possible cancer symptoms ($b = -0.10$ [−0.19, −0.02], $p = 0.016$) (Supplementary Note 1, Table S5a). We also detected a significant positive relationship between the dichotomous variable (change vs. no change) and GPs' general attitudes towards cancer risk calculators ($OR = 1.17$ [1.03, 1.33], $p = 0.015$) (Supplementary Note 1, Table S5b).

**Algorithm impact on decision appropriateness**. We categorised 63% of initial referral decisions and 68% of final referral decisions

**Table 4 Changes in inclination to refer, risk estimates and QCancer.**

| Changes in inclination to refer | | Risk estimate pre-algorithm | QCancer risk score | Risk estimate post-algorithm |
|---|---|---|---|---|
| Towards referral | 327 (40.5%) | 12.4% (13.0) | 31.2% (22.6) | 27.5% (20.1) |
| Away from referral | 481 (59.5%) | 22.6% (20.0) | 3.5% (5.4) | 7.6% (9.5) |
| Total | 808 (100%) | | | |

Changes in the inclination to refer post-algorithm either towards or away from referral and associated means (SD) of GPs' pre- and post-algorithm risk estimates and means (SD) of the QCancer risk score.

**Table 5 Decision appropriateness.**

| | 'Uncertain' responses excluded from count | | 'Uncertain' responses classed as inappropriate | |
|---|---|---|---|---|
| | Pre-algorithm | Post-algorithm | Pre-algorithm | Post-algorithm |
| Appropriate | 1925 (75.2%) | 1984 (77.5%) | 1982 (63.1%) | 2147 (68.4%) |
| Inappropriate | 635 (24.8%) | 576 (22.5%) | 1158 (36.9%) | 993 (31.6%) |
| Total | 2560 (100%) | 2560 (100%) | 3140 (100%) | 3140 (100%) |

Frequency of appropriate and inappropriate referral decisions before and after seeing the algorithm, with 'uncertain' responses first excluded and then included in the count as 'inappropriate'.

as appropriate; 'uncertain' responses were classed as inappropriate, in the first instance. GPs' decisions post-algorithm were significantly more appropriate than pre-algorithm (OR = 1.45 [1.27, 1.65], $p < 0.001$) (Supplementary Note 1, Table S6a). When we excluded from the count instances where either pre- or post-algorithm decisions were 'uncertain', the results were similar (OR = 1.26 [1.06, 1.50], $p < 0.001$) (Supplementary Note 1, Table S6b). Table 5 presents the frequencies. Thus, the odds of an appropriate referral decision were between 1.3 and 1.5 times higher after respondents received the algorithm's risk estimates.

**Learning**. We explored changes in the GPs' initial risk estimates over time and observed the following:

1. The mean absolute difference between initial risk estimates and QCancer was significantly smaller in the second session: 17.3% (SD 16.83) in the first session vs. 15.70% (SD 15.91) in the second session. This difference between sessions was significant ($b = -1.63\%$ [−2.53, − 0.72] $p < 0.001$) (Supplementary Note 1, Table S7). Thus, during the second session, GPs closed the gap between their initial estimates and QCancer by 1.6 percentage points on average.

2. There was a significant negative relationship between vignette order and the absolute difference between initial risk estimates and QCancer ($b = -0.14$ [−0.22, −0.06], $p < 0.001$) (Supplementary Note 1, Table S8), indicating that GPs' estimates improved over time (moved closer to the algorithm). Figure 1 demonstrates this trend. The pattern of the results suggests that the improvement was not continuous but occurred mainly in the second session (vignette order 11 to 20). Furthermore, several GPs wrote comments that suggested awareness that learning had taken place, either in general or in relation to how specific symptoms contributed to a patient's risk of colorectal cancer (Supplementary Note 2).

**Algorithm disposition questionnaire (ADQ)**. At the end of the study, GPs expressed a generally positive attitude towards the algorithm: the mean ADQ score was 5.11 (7-point response scale from strongly disagree (1) to strongly agree (7)) (SD 1.13, *Median* 5.29). When we explored possible predictors of ADQ, we detected significant relationships with gender (male = 0, female = 1): $b = -0.37$ [−0.73, −0.02] $p = 0.040$; general attitude towards cancer risk calculators ($b = 0.22$ [0.10, 0.33] $p < 0.001$);

confidence when assessing patients with possible cancer symptoms ($b = -0.54$ [−0.91, −0.18] $p = 0.004$), and algorithm information: $b = 0.36$ [0.01, 0.70] $p = 0.043$.

**Discussion**

Our study findings provide insights into the potential impact and benefits of using cancer risk calculators in clinical practice for the earlier detection of cancer. In general, changes in inclination to refer were consistent with changes in risk estimates. Importantly, we measured an improvement in the appropriateness of referral decisions post-algorithm. Although this improvement was small, it was statistically significant. There are several reasons for the small effect of the algorithm on referral decisions. First, decisions are notoriously difficult to change, despite changes in associated risk estimates[20]. Decisions may be based on other factors in addition to risk estimates[21,22]. For example, we found some evidence that both general attitudes towards cancer risk calculators and confidence in assessing patients with possible cancer symptoms were related to shifts in inclination to refer post-algorithm. Second, GPs abided by the precautionary principle: although around half of the vignettes had risk ≤3%, not necessitating referral, referrals post-algorithm remained high at 68%. Referral inclination changed in accordance with the algorithm more if GPs had underestimated than overestimated risk. In other words, GPs erred on the side of caution and were disinclined to change an initial referral decision, even if the algorithm indicated that referral was not required. Third, it is possible that were GPs not required to provide explicit risk estimates first, and thus, an anchor for their decisions, they would have relied more on the algorithm, which could have exerted a greater influence on decisions. Finally, decision appropriateness started from a relatively high baseline. In this simulated environment, GPs generally made appropriate referral decisions pre-algorithm. Still, a 5% absolute improvement post-algorithm could translate into cancers diagnosed earlier, as well as unnecessary referrals being avoided.

GPs overestimated cancer risk most of the time. This may be because they do not regularly provide explicit risk estimates in the form of numerical probabilities. Risk estimates often remain implicit and GPs do not receive any systematic feedback that would enable them to calibrate their estimates better. Ours is not the only study that identified poor calibration of risk estimates. A survey of Canadian GPs found that they overestimated the absolute 8-year cardiovascular disease risk for two hypothetical

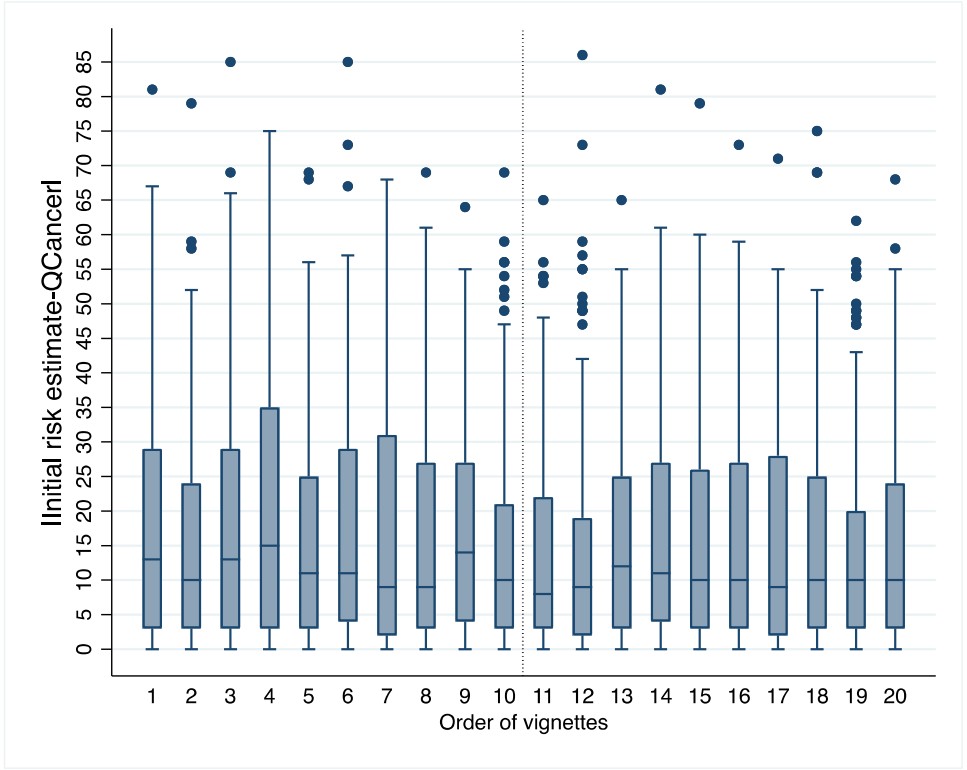

**Fig. 1 A learning effect.** Box plot depicting the absolute difference between GPs' initial risk estimates and QCancer scores by order of vignette presentation. The box plot shows the median, interquartile range, minimum and maximum values, and outliers for each vignette order. The dotted vertical line indicates the start of the second session (vignette order 11), when the pattern appears to stabilise, suggesting learning consolidation. $n = 3140$ data points (157 GPs x 20 vignettes). The source dataset for the Figure can be found in Supplementary Data 2.

patients[23]. A recent study found that US clinicians at outpatient clinics overestimated the probability of disease for four scenarios common in primary care and that overestimation persisted after receiving both positive and negative test results[24]. Focusing on single disorders in the absence of a differential in the above studies is likely to have contributed to overestimation. Consideration of alternative diagnostic possibilities is one strategy to drive down overestimation[25].

An unexpected but encouraging finding from our study was the learning effect, which became apparent during the second session: GPs became better calibrated, i.e., their initial risk estimates moved closer to the algorithm. This suggests that GPs were noticing and learning the probabilistic relationships between single or multiple symptoms and algorithmic risk (covariation learning). GPs were aware of this learning, as is evident in their comments. The finding that improvement occurred in the second session is consistent with the literature on learning consolidation, i.e., the stabilisation of memory traces after their initial acquisition[26]. Consolidation of learning happens during off-line periods, when participants are not engaged with the task at hand. A fruitful avenue for future research would therefore be to explore the learning that occurs during repeated trials with algorithm feedback, and the factors affecting consolidation, such as the time interval between training sessions, and the frequency and spacing of booster training sessions. Risk calculators may have a role as training tools, enabling GPs to internalise the weighting of risk factors such as age, smoking, alcohol intake, family history, and specific symptoms on the risk of undiagnosed cancer in patients. GPs who participated in our study frequently commented on the opportunity for reflection that the study provided. Opportunities for reflection, learning and improvement

could engender positives attitudes towards risk calculators and commitment to their future use.

We found moderately high awareness and a generally positive attitude towards cancer risk calculators amongst participants at the start of the study. Nevertheless, the reported availability in clinical practice and the self-reported use of these calculators was low. Our study confirms that cancer risk calculators remain an underused resource in UK primary care. A cross-sectional postal survey of UK GPs in 2017 also found low awareness, availability and use of cancer risk calculators[15]. However, the positive comments about the algorithm made by our participants at the end of the study suggest that barriers to the adoption of cancer risk calculators are most likely practical, such as a lack of supportive activities during their introduction[27] and a lack of seamless integration into the clinical workflow[14]. External support from cancer networks in the form of webcasts, email updates and newsletters were found to improve the acceptance and use of RATs[27]. Therefore, a combination of external support and training sessions may help to increase the uptake of cancer risk calculators in the future.

Our study also aimed to investigate whether informing GPs of the algorithm's provenance, validation and accuracy increases their willingness to change their decisions in accordance with the algorithm. We did not find evidence for this. We did however find that GPs who received information about the algorithm expressed more positive attitudes towards it at the end of the study. We therefore recommend that risk algorithms are always introduced with care in clinical practice, ensuring that users have all the necessary information about them. We are currently investigating different ways of introducing the algorithm to users and explaining how the risk estimates were derived, with a view to

increasing transparency, explainability and trust, and exploring effects on learning.

We did not mention the 3% risk referral threshold set by NICE to our participants and do not know how many were aware of it. Of the 94 GPs who wrote free-text comments at the end of the study (i.e., 60% of the sample), only four mentioned it. One GP wondered if a cut-off had been established for the risk threshold. Based on these comments, we do not expect that the 3% threshold is widely known among GPs. The NICE guidelines are procedural, based on age cut-offs, and do not make explicit reference to a numeric threshold. Thus, GPs currently base 2WW referrals on these procedural guidelines and/or their own assessment of risk, which may or may not be informed by a cancer risk calculator. To support the earlier diagnosis of cancer and avoid unnecessary referrals, a better integration of these two approaches is required, while more research is needed to establish which approach is more acceptable to GPs and can lead to better targeted referrals.

One limitation of the study concerns our sample, which may not be representative of the UK GP population in terms of age and gender. To our knowledge, there are no such statistics publicly available, though it is known that female GPs (53.5% in our sample) have been outnumbering male GPs since 2014[28]. GPs self-selected into the study. Consequently, the topic may have attracted predominantly those interested in cancer risk algorithms. Nevertheless, only a minority of participants had such tools available at their practice (30%), and of these, 40% never used them. Thus, we do not expect that experience with such tools biased the results in any systematic fashion. Furthermore, the incentives (both monetary and the completion certificate/ personalised feedback for CPD) were designed to increase the sample's representativeness by making the study attractive to GPs who did not necessarily have a special interest in algorithmic tools.

To study the impact of the algorithm on clinical judgements and decisions, we created a controlled environment quite different from real-life clinical consultations. Our participants saw a series of patient descriptions in sequence, all containing features and risk factors associated with colorectal cancer, and they were specifically asked to consider the possibility of colorectal cancer. In practice, these patients are unlikely to present one after the other or on the same day. They may present with a multitude of vague symptoms that makes their diagnosis and management more uncertain. GPs may need to deal with external pressures, such as patient expectations, or practice policies to increase or reduce cancer referrals. They will not be asked to provide explicit risk estimates, neither will they make a decision about referral on a 1–5 response scale. Mindful that our participants would sometimes need more information or a further consultation before making a decision, we opted for a response scale that measured inclination rather than a final decision and offered them the option to remain 'uncertain'. There are thus several differences between our simulated computer task and dealing with real people at a busy practice. We had to control the amount, type and format of information provided to participants to ensure standardisation and remove confounders. We do not claim that our GPs would respond in the same way to these patients if they saw them at their practice—even though, there is evidence that responding to written clinical vignettes provides a good approximation of real-life behaviour[29]. Our aim was to determine to what extent GPs are willing and able to integrate estimates from a cancer risk calculator into their own intuitive estimates and if such calculators have the potential to improve referral decisions. Our findings provide a positive answer to both questions.

In summary, there is value in the use of cancer risk calculators in clinical practice, but they are currently underused. Their potential role as resources for training should be further explored

and they could become part of training materials for GP trainees and new GP starters. The desired result would be a better uptake of these tools, as well as a greater understanding of the weighting of risk factors and symptoms when assessing patients, with the ultimate aim to improve the early diagnosis of cancer.

## Data availability

The data that support the findings of this study are available in Dryad with the identifier https://doi.org/10.5061/dryad.76hdr7swm[30]. The dataset includes both the raw and calculated data that support Fig. 1.

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

## Acknowledgements

The study was funded by a Cancer Research UK grant awarded to Olga Kostopoulou. Funding Scheme: Population Research Committee - Project Award, Reference A28634. The sponsor had no role in any stage of the study, including in study design, data analysis or manuscript preparation.

## Author contributions

O.K. conceived the study, obtained the funding, and acted as project leader. All authors contributed to the original idea for the study. O.K. and K.A. drafted the paper. K.A. adapted and created clinical vignettes with input from O.K. and B.P. B.P. programmed the vignette survey on the Qualtrics platform and collected the data. O.K. and B.P. conducted the statistical analyses. All authors discussed the content and contributed to editing the manuscript.

## Competing interests

The authors declare no competing interests.
