## [Peer Review File · Communications Medicine]

Reviewers' comments:

Reviewer #1 (Remarks to the Author):

This is a very interesting paper, with a novel study design, large numbers of vignettes and GP participants, investigating how cancer risk algorithms influence (English) GPs' clinical risk assessments and referral decisions of colorectal cancer. The authors provided data for transparent and reproducible research, which is very good. Some parts of the study are well-discussed, like the learning effect.

Below are a few suggestions for the authors to consider:

P2 Abstract "Decisions became significantly more appropriate". What did it mean of "more appropriate"? More appropriate in what way? Please make it clearer.

P7 Result. The GP characteristics should be in a table, not just described in text – this is important information. It would be helpful to know how representative this GP sample is, relative to the "GP lists" owned by the lead author, or compared with the GP characteristics of those with MRCGP (if such info is available). It would be helpful for the author to discuss whether and to what extent the incentives influence GP's willingness to participate in this study, and whether the incentive influence sample representativeness.

One of the key questions is whether the GPs participating in this study are aware of the referral threshold set by the NICE guideline – knowing this could influence their decision on referral. Not sure whether the authors included this question in their study. This is a point worth explanation and discussion.

The authors mentioned this paper is the first of a planned series of studies, then what is the next? It is good to mention in the discussion section to inform the readers. For example, any other (qualitative/quantitative) work to complement the submitted work?

The authors discussed the potential of risk prediction tools "become part of training materials for GP trainees and new GP starters" Any implications in education for medical school students (MBBS/MBChB programme)?

Tables: the current tables are too descriptive. The authors may consider restructuring the tables to highlight the important information of the paper. The current information in the table is difficult for the readers to follow and understand the gist of the paper. Suggest combining some tables to reduce redundancy (e.g. Table 1 and Table 2). Some tables may not be necessary. Include GP characteristics in the table. The regression analysis in Appendix 4 is an important part of the analysis and the study and should be in the main text, in line with the description in the Result section. The random effect of GP is of interest and should be reported.

Figure: not sure what Figure 1 adds to the paper. It is understandable of the learning effect during this exercise. Not sure whether the authors need this figure to illustrate the point they tried to make on P10.

Reviewer #2 (Remarks to the Author):

This vignette study explores the impact on referral decisions of providing GPs with cancer risk estimates derived from a diagnostic risk prediction model. GPs were asked to estimate cancer risk and indicate the likelihood that they would refer via the 2 week wait pathway before and after being shown the model risk estimate. The study found that provision of model risk estimates led to a proportion of GPs (12% of cases) changing their referral decision and that it led to a small improvement in the 'appropriateness' of referral decisions. This findings indicates that risk prediction tools might improve referral decisions if used within primary care. The study also found that GPs risk estimates became more closely aligned to the those of the algorithm during the course of the study - this is intriguing as it highlights a potential role for risk models as educational tools.

I found the manuscript well written and very easy to follow. The methods are well described and the statistical approaches appear broadly appropriate. The results are clearly presented and appendices comprehensive. The discussion is balanced and highlights the limitations of the work, most notably that the study cannot fully mirror standard clinical practice/decision making.

This study makes a significant contribution to the literature - much research time and resources are directed towards the development and validation of cancer risk tools but (as the authors highlight) they are underused. This study improves our understanding of the possible impact of cancer prediction tools on GP referral decisions and highlights a possible use for them as educational tools.

I only have a couple of minor suggestions which the authors may wish to consider:

- 1) Can the authors give their rationale for choosing QCancer over the other models e.g. eRATS? Because QCancer is the most common? It may also be helpful for readers (particularly those outside the UK) to provide a line or two of background on QCancer.
- 2) The authors highlight that one potential reason for a relatively small change in referral decisions was that decisions are hard to change. I wonder whether the authors think that if the tool were to be used within clinical practice as part of the initial decision making process (as the model developers intend) it might have more impact on referral decision making? I.E might the study underestimate the influence of providing risk estimates due to its design?

Garth Funston

Rebuttal letter

We would like to thank both reviewers for their positive feedback and hope that we have responded to all their comments and suggestions.

Reviewer #1

This is a very interesting paper, with a novel study design, large numbers of vignettes and GP participants, investigating how cancer risk algorithms influence (English) GPs' clinical risk assessments and referral decisions of colorectal cancer. The authors provided data for transparent and reproducible research, which is very good. Some parts of the study are well-discussed, like the learning effect. Below are a few suggestions for the authors to consider:

Comment: P2 Abstract "Decisions became significantly more appropriate". What did it mean of "more appropriate"? More appropriate in what way? Please make it clearer.

Response: We have now changed this sentence to: "Decisions became more consistent with the NICE 3% referral threshold"

Comment: P7 Result. The GP characteristics should be in a table, not just described in text – this is important information. It would be helpful to know how representative this GP sample is, relative to the "GP lists" owned by the lead author or compared with the GP characteristics of those with MRCGP (if such info is available). It would be helpful for the author to discuss whether and to what extent the incentives influence GP's willingness to participate in this study, and whether the incentive influence sample representativeness.

Response: We kept the GP characteristics in the main text rather than a table, since we only requested age, gender, and years of experience. Below, we include such a table. If the Editor thinks that the presentation is better as a table, we can include it and remove the information from the main text. This would increase the number of tables.

	Fully qualified GPs		GP trainees		Total
	Females (N = 79)	Males (N = 71)	Females (N = 5)	Males (N = 2)	
Mean age in years (SD)	43.5 (7.4)	45.7 (9.5)	31.0 (2.3)	32.5 (3.5)	44 (8.7)
Mean experience in years (SD)	13.2 (8.4)	15.1 (9.9)	0	0	14 (9.0)

We cannot make comparisons with a larger GP population: to our knowledge, there are no publicly available age and gender statistics for the UK GP population; furthermore, we do not keep any information on our list beyond the GPs' e-mail addresses. We have, however, added a paragraph in the discussion, following the editor's suggestion, which discusses the possible lack of representativeness as a limitation (see Discussion, page 12: "One limitation of the study..."). There, we also discuss how the incentives may have influenced sample representativeness.

Comment: One of the key questions is whether the GPs participating in this study are aware of the referral threshold set by the NICE guideline – knowing this could influence their decision on referral. Not sure whether the authors included this question in their study. This is a point worth explanation and discussion.

Response: This is a very good point. We have now added a new paragraph in the discussion about this (p. 12), starting with “We did not mention the 3% risk referral threshold...”.

Comment: The authors mentioned this paper is the first of a planned series of studies, then what is the next? It is good to mention in the discussion section to inform the readers. For example, any other (qualitative/quantitative) work to complement the submitted work?

Response: Thank you for this suggestion. We have included this in the discussion: “We are currently investigating different ways of introducing the algorithm to users and explaining how the risk estimates were derived, with a view to increasing transparency, explainability and trust, and exploring effects on learning.” (p. 12)

Comment: The authors discussed the potential of risk prediction tools “become part of training materials for GP trainees and new GP starters” Any implications in education for medical school students (MBBS/MBChB programme)?

Response: This is an excellent suggestion, and we were indeed planning to explore it as part of a medical undergraduate student project starting early next year. We have not however added anything to the manuscript, since it is somewhat peripheral to its main focus on cancer referrals by GPs.

Comment: Tables: the current tables are too descriptive. The authors may consider restructuring the tables to highlight the important information of the paper. The current information in the table is difficult for the readers to follow and understand the gist of the paper. Suggest combining some tables to reduce redundancy (e.g. Table 1 and Table 2). Some tables may not be necessary. Include GP characteristics in the table.

Response: To improve readability of Tables 1 and 2, we have made the column headings more descriptive. However, merging them would result in a long table of 22 rows, presenting frequency of use calculator (never, sometimes, always) separately for each type of cancer risk calculator. Furthermore, this merged table could not include all the information, e.g., means for attitudes per frequency of use that Table 2 currently presents. It would also have several empty cells for the less common types of calculator. Therefore, we think that the tables should be kept separate, because they are simpler and clearer than a combined table. We responded earlier to the comment about presenting GP characteristics in a table.

Comment: The regression analysis in Appendix 4 is an important part of the analysis and the study and should be in the main text, in line with the description in the Result section. The random effect of GP is of interest and should be reported.

Response: We are now reporting all random effects in the regression tables and indicate in the text which Appendix table supports the results. As stated on p. 6 of the manuscript, “The regression tables are presented in Appendix 3 (*formerly Appendix 4*), in the sequence that they appear in the text.” We have kept all 12 tables in Appendix 3, since the journal may have specific requirements about where tables should be placed.

Comment: Figure: not sure what Figure 1 adds to the paper. It is understandable of the learning effect during this exercise. Not sure whether the authors need this figure to illustrate the point they tried to make on P10.

Response: Figure 1 illustrates the point about learning consolidation. As we state on p. 10, “The pattern of the results suggests that the improvement was not continuous but occurred

mainly in the second session (vignette order 11 to 20).” We have now added a vertical line on the figure to show where the improvement appears to stabilise and have included an explanation in the legend of the Figure: “The dotted vertical line indicates the start of the second session, when the pattern of improvement in the risk estimates stabilised, suggesting learning consolidation.”

Reviewer #2 (Garth Funston)

This vignette study explores the impact on referral decisions of providing GPs with cancer risk estimates derived from a diagnostic risk prediction model. GPs were asked to estimate cancer risk and indicate the likelihood that they would refer via the 2 week wait pathway before and after being shown the model risk estimate. The study found that provision of model risk estimates led to a proportion of GPs (12% of cases) changing their referral decision and that it led to a small improvement in the 'appropriateness' of referral decisions. This findings indicates that risk prediction tools might improve referral decisions if used within primary care. The study also found that GPs risk estimates became more closely aligned to the those of the algorithm during the course of the study - this is intriguing as it highlights a potential role for risk models as educational tools.

I found the manuscript well written and very easy to follow. The methods are well described and the statistical approaches appear broadly appropriate. The results are clearly presented and appendices comprehensive. The discussion is balanced and highlights the limitations of the work, most notably that the study cannot fully mirror standard clinical practice/decision making.

This study makes a significant contribution to the literature - much research time and resources are directed towards the development and validation of cancer risk tools but (as the authors highlight) they are underused. This study improves our understanding of the possible impact of cancer prediction tools on GP referral decisions and highlights a possible use for them as educational tools.

I only have a couple of minor suggestions which the authors may wish to consider:

1) Can the authors give their rationale for choosing QCancer over the other models e.g., eRATS? Because QCancer is the most common? It may also be helpful for readers (particularly those outside the UK) to provide a line or two of background on QCancer.

Response: We chose QCancer only because it is publicly available, as is the underlying computer code. We now state this in the manuscript (p. 4). By visiting the website, also provided on p. 4, readers can experience the algorithm by inputting symptoms and can find the evidence on which the algorithm is based. The algorithm description (Appendix 2, Box 1) describes QCancer (without naming it).

2) The authors highlight that one potential reason for a relatively small change in referral decisions was that decisions are hard to change. I wonder whether the authors think that if the tool were to be used within clinical practice as part of the initial decision making process (as the model developers intend) it might have more impact on referral decision making? I.E might the study underestimate the influence of providing risk estimates due to its design?

Response: We agree and have added a relevant sentence to the discussion (p. 11).

REVIEWERS' COMMENTS:

Reviewer #1 (Remarks to the Author):

The revised discussion section is strengthened, particularly the paragraphs discussing GP's awareness of the threshold, sample representativeness, and the influence of incentives. These are the improvement of the paper.

Another improvement is in the Result section – signposting the readers where to find more details of statistical results in the Appendix.

One minor suggestion: if the authors prefer to include Figure 1, perhaps change the labels of the y-axis to make it more understandable to the readers. The label for the x-axis could be changed to “The order of vignettes”.

Reviewer #2 (Remarks to the Author):

Thanks for the opportunity to take another look at this interesting paper. The authors have fully addressed my minor comments in their revision. Garth Funston

Reviewer #1 (Remarks to the Author):

The revised discussion section is strengthened, particularly the paragraphs discussing GP's awareness of the threshold, sample representativeness, and the influence of incentives. These are the improvement of the paper.

Another improvement is in the Result section – signposting the readers where to find more details of statistical results in the Appendix.

One minor suggestion: if the authors prefer to include Figure 1, perhaps change the labels of the y-axis to make it more understandable to the readers. The label for the x-axis could be changed to “The order of vignettes”.

RESPONSE: We have changed the label of the Y axis to: |Initial risk estimate-QCancer|. As the figure legend states, this is the absolute difference between GPs' initial risk estimates and QCancer scores. We have also changed the label for the X axis according to the reviewer's suggestion.

Reviewer #2 (Remarks to the Author):

Thanks for the opportunity to take another look at this interesting paper. The authors have fully addressed my minor comments in their revision. Garth Funston